# Dynamic Light Scattering Plus Scanning Electron Microscopy: Usefulness and Limitations of a Simplified Estimation of Nanocellulose Dimensions

**DOI:** 10.3390/nano12234288

**Published:** 2022-12-02

**Authors:** Quim Tarrés, Roberto Aguado, Justin O. Zoppe, Pere Mutjé, Núria Fiol, Marc Delgado-Aguilar

**Affiliations:** 1LEPAMAP-PRODIS Research Group, University of Girona, C/ Maria Aurèlia Capmany, 61, 17003 Girona, Spain; 2Department of Chemical and Agricultural Engineering and Agrifood Technology, University of Girona, C/Maria Aurèlia Capmany, 61, 17003 Girona, Spain; 3Department of Materials Science and Engineering, Universitat Politecnica de Catalunya (UPC), 08019 Barcelona, Spain

**Keywords:** cellulose nanocrystals, cellulose nanofibers, dynamic light scattering, hydrodynamic diameter, nanocellulose, scanning electron microscopy

## Abstract

Measurements of nanocellulose size usually demand very high-resolution techniques and tedious image processing, mainly in what pertains to the length of nanofibers. Aiming to ease the process, this work assesses a relatively simple method to estimate the dimensions of nanocellulose particles with an aspect ratio greater than 1. Nanocellulose suspensions, both as nanofibers and as nanocrystals, are subjected to dynamic light scattering (DLS) and to field-emission scanning electron microscopy (FE-SEM). The former provides the hydrodynamic diameter, as long as the scatter angle and the consistency are adequate. Assays with different angles and concentrations compel us to recommend forward scattering (12.8°) and concentrations around 0.05–0.10 wt %. Then, FE-SEM with magnifications of ×5000–×20,000 generally suffices to obtain an acceptable approximation for the actual diameter, at least for bundles. Finally, length can be estimated by a simple geometric relationship. Regardless of whether they are collected from FE-SEM or DLS, size distributions are generally skewed to lower diameters. Width distributions from FE-SEM, in particular, are well fitted to log-normal functions. Overall, while this method is not valid for the thinnest fibrils or for single, small nanocrystals, it can be useful in lieu of very high-resolution techniques.

## 1. Introduction

Measuring non-spherical polydisperse nanomaterials in an accurate, reproducible, and straightforward way remains a challenge. This is the case for cellulose nanofibers (CNFs) and, to a lesser extent, cellulose nanocrystals (CNCs) [1,2]. The former are present in aqueous suspensions as networks of entangled fibrils with a high aspect ratio (length/diameter), usually between 20 and 200 [3,4]. CNF entanglement hinders the determination of their length even by the most advanced microscopy techniques, as distinguishing both ends of each individual fibril is often impossible. Therefore, researchers often resort to microscopy for width measurements, possibly aided by software tools such as ImageJ or DiameterJ [5,6]. Even for this task, most works rely on very high-resolution microscopes. For instance, Campano et al. [7] opted for transmission electron microscopy and image processing when measuring the size of CNCs and CNFs. Another option is presented by scanning electron microscopy (SEM) [8], but electron beams may damage CNFs at high magnifications, and staining is often necessary to obtain higher contrast. Furthermore, in an approach that inspired the present work, Gamelas et al. [2] used atomic force microscopy for measuring the width of CNFs, and their length was estimated with the aid of dynamic light scattering (DLS).

Indeed, DLS is a non-destructive, extremely popular technique to measure particle size [9,10,11]. It has been satisfactorily used to measure the size of CNCs in multiple reports [11,12,13,14]. However, we must display a critical approach when addressing DLS for CNC dimensions. First, since CNCs are non-spherical particles whose aspect ratio may be around 20 [15], the hydrodynamic diameter reported by DLS instruments should not be confused with their size (vaguely), with their length, or with the actual mean diameter. Second, unless CNCs present high surface charge (e.g., due to –O–SO_3_– moieties) or are stabilized by another component besides them and water, the detection of single crystals is not easily distinguished from that of their aggregates [14]. Third, the choice of the scatter angle and the concentration of the sample are seldom considered, even though they are of utmost importance when dealing with polydisperse samples that tend to agglomerate [16]. Interestingly, Boluk and Danuma [17] also supported microscopy with DLS to measure CNCs, while not relying on the reported hydrodynamic diameter, but on the diffusion coefficient.

Although the user must always be critical of the output of DLS software, recent devices tend to offer straightforward measurements and user-friendly procedures [18,19]. Likewise, tabletop SEM instruments, usually more user-friendly and less expensive than full-sized instruments, are progressively improving their resolution [20,21]. We aim to take advantage of the advances in both fields. In this context, this work explores the possibility of using a user-friendly, straightforward DLS-based instrument, along with SEM with relatively low magnification (×20,000 at most), for estimating the dimensions of both CNCs and CNFs. We begin by assessing the effect of scattering angles and sample concentration on DLS reports. SEM images of conventionally prepared CNCs and oxycellulose nanofibers allow us to display diameter distributions. Then, the resulting average diameter is geometrically related to the measurements of the hydrodynamic diameter that were deemed reliable. Finally, commonly reported properties of nanocellulose are discussed in relation to its dimensions.

## 2. Materials and Methods

### 2.1. Materials

Bleached eucalyptus kraft pulp (BEKP), unrefined (15 °SR), was provided by Ence (Navia, Spain) and used to produce CNFs. In turn, CNCs were obtained from a cotton-based, ashless Whatman filter paper, #42, purchased from Sigma-Aldrich (Schnelldorf, Germany).

2,2,6,6-tetramethylpiperidine-1-oxyl radical (TEMPO), NaClO, NaOH, copper(II) ethylenediamine (Cuen), and poly(diallyldimethylammonium chloride) (PDADMAC) of medium molecular weight were purchased from Sigma-Aldrich (Sigma-Aldrich Química SA, Barcelona, Spain). NaBr and concentrated H_2_SO_4_ (97 wt %) were received from Scharlab (Scharlab SL, Sentmenat, Spain). Sodium polyethylene sulfonate (PES-Na) was provided by BTG (BTG Instruments GmbH, Weßling, Germany), along with the surface charge analyzer.

### 2.2. Production of CNFs

The regioselective oxidation of BEKP took place at 1% consistency and at pH 10, as described elsewhere [22,23]. Briefly, NaBr (3 g) and TEMPO (0.06 g) were mixed with BEKP (30 g) in aqueous medium, and then a certain amount of NaClO was added. A 0.5 M NaOH solution was used to adjust the pH. After the reaction, the oxidized pulp was washed with distilled water and had its consistency adjusted to 1% again.

The fibrillation process was carried out in a high-pressure homogenizer (HPH), NS1001L PANDA 2 K-GEA. A suspension of oxidized fibers was passed 3 times at 300 bar and 4 times at 600 bar. CNFs produced through this method are coded from now on as CNFs5, CNFs10, and CNFs15, where the number signifies the ratio of NaClO to pulp used for the regioselective oxidation (5, 10, or 15 mmol/g, respectively).

### 2.3. Production of CNCs

First, 7 g of filter paper was placed in a borosilicate glass beaker containing 150 mL of H_2_SO_4_ 65 wt %. The suspension was kept at 50–55 °C and under agitation by means of a heating plate and a Teflon-coated magnet. After 30 min, 100 g of ice was added to stop the reaction. Nanocrystals were sedimented by three consecutive centrifugations (10,000× *g*, 15 min), discarding the free acid solution and adding distilled water after each of them. Then, the suspension of CNCs was wrapped in a nanofiltration membrane with a molecular cutoff of 10 kDa. The resulting bags were immersed in distilled water for 7 days, changing the washing water every two days, to remove the remnants of acid. Neutralized CNCs (pH 6–7) were stored in topaz bottles and under 4 °C.

### 2.4. Other Characterization Techniques

The transmittance at 600 nm of 0.1 wt % suspensions was measured by means of a Shimadzu spectrophotometer, model UV-1201. The weight fraction of a 0.2 wt % suspension, on a dry basis, that did not settle after centrifugation for 20 min at 1254× *g*, was regarded as the yield of nanofibrillation [24,25].

The water retention value (*WRV*) of nanocellulose was also determined gravimetrically. For that, CNF hydrogels containing excess water were placed in containers with a nitrocellulose membrane whose pore size was 0.22 µm. Containers were then centrifuged at 3000× *g* for 15 min. The filter cake was collected, weighed (*m_W_*), oven-dried at 105 °C, and weighed again (*m_D_*). *WRV* was calculated from Equation (1):(1)$$WRV=\frac{m_{W}-m_{D}}{m_{D}}$$

Charge density (*CD*) was estimated by potentiometric back titration [26]. Approximately 0.05 g of nanocellulose was suspended in 50 mL of PDADMAC 0.001 N. Then, the excess PDADMAC was titrated with *PES-Na* 0.001 N until the isoelectric point (0 mV) was reached. *CD* is given by:(2)$$CD=\frac{\left(V-V_{b}\right)\;c_{PES-Na}}{m}$$
where *V* is the volume of the titrating agent, *c_PES-Na_* is its concentration, *V_b_* is the volume of the titrating agent spent in a blank experiment (without nanocellulose), and *m* is the weight of the sample on a dry basis.

The degree of polymerization (DP) was estimated as detailed elsewhere [27]. Briefly, we dissolved the cellulosic material in Cuen (copper (II) ethylenediamine complex) at different concentrations and measured the intrinsic viscosity by means of a capillary viscometer. Then, the average molecular weight was calculated from the Mark–Houwink equation [28] and divided by the weighted average molecular weight, considering both substituted and unsubstituted anhydroglucose units.

X-ray diffraction patterns of dry samples were performed with Cu-Kα radiation (154.18 pm, 40 mA) by means of Bruker’s diffractometer D8 Advance. The intensity corresponding to the minimum at 2θ ~ 18.5° and the intensity for the (200) peak (at 2θ ~ 22°) were computed to calculate the crystallinity index, following the Segal method [29].

### 2.5. Dynamic Light Scattering

Before DLS measurements, nanocellulose samples were diluted and dispersed by means of an UltraTurrax device, IKA T25 IKA T25 (IKA®-Werke GmbH, Staufen, Germany). Suspended bubbles were removed by ultrasonication. Then, the hydrodynamic diameter of CNFs and CNCs was estimated by using a ZetaSizer device, model Nano-ZS (Malvern Panalytical Ltd., Malvern, UK). Measurements took place at pH 6–8 and without additional ions, other than Na+ as counter-ions for the carboxylate and sulfate groups. Preliminary tests were performed to find out appropriate ranges for the testing conditions. Hence, samples of different concentrations, from 0.0001 wt % to 0.1 wt %, were placed in polystyrene cuvettes. Besides the concentration, the relevance of the scatter angle was assessed as well, performing tests at 12.8° (forward scattering) and 173° (backscattering). In each case, the distribution of particle size by intensity was collected.

### 2.6. Estimation of Particle Dimensions

Field-emission scanning electron microscopy (FE-SEM) was performed by means of a ZEISS DSM 960A device (ZEISS Iberia, Madrid, Spain), using carbon coating, a secondary electron detector, and a voltage of 7 kV. Micrographs were processed by the ImageJ software package as described in a previous work [6]: conversion into 8-bit images, bandpass filter, background subtraction, *Close* filter. Dimensional data were gathered by means of the same software and the *Fractal Box Count* plugin. After at least 300 counts for each sample, a histogram of the diameters was produced. We carried out fittings to Gaussian, Lorentzian, and log-normal distribution functions with OriginLab’s software package OriginPro 8.5.

CNFs and CNCs were modeled as cylinders whose volume equals the volume of the sphere predicted by DLS. In other words, their length (*L*) can be estimated from:(3)$$\frac{d_{H}{}^{3}\;\pi }{6}=\frac{L\;\pi \;d_{m}{}^{2}}{4}$$

In Equation (3), *d_m_* is the diameter of the fibrils as measured from FE-SEM images, and *d_H_* is the hydrodynamic diameter reported from DLS assays.

## 3. Results and Discussion

### 3.1. Definition of Proper Testing Conditions

Consistently, for all samples of nanocellulose, forward scattering (12.8°) was more reproducible, reliable, and convenient than backscattering (173°). Figure 1 shows that when processing CNF samples with a backscatter configuration, there are unexpected discontinuities, as if different populations coexisted. Possibly, the (backward) scattering of nanofibers at high angles could be mistaken for that of elementary fibrils. As hinted for other kinds of particles, small angles can exclude the parasitic effect of the rotational motion of non-spherical particles [16]. Moreover, high angles often produced bimodal correlation functions in the case of CNF samples, high polydispersity indices (0.7–1), and lower reproducibility. Regarding the latter aspect, it should be mentioned that the distribution of the CNFs obtained after oxidation with 10 mmol of NaClO per gram (CNFs10), as displayed in Figure 1, was not replicated by other runs with the same sample.

Regarding CNCs, both forward and backward scattering often reported one population. However, some backscattering runs of the same sample, including the one shown in Figure 1, unexpectedly detected large particles, up to 5.5 µm. Polydispersity was not significantly different. In fact, a scatter angle of 173° has been successfully used for CNCs [17]. However, taking into account the ulterior comparison with micrographs, we opted for small scatter angles, not only for CNFs (for which the decision is clear) but also for CNCs.

When it comes to the effects of concentration, Figure 2 presents the intensity-weighted average values for the hydrodynamic diameter, obtained for consistencies not higher than 0.5 wt %. Each of the points is the arithmetic mean of at least three intensity-weighted average *d_H_* values.

Optimally, the size distribution does not depend on the concentration of the sample. Otherwise, there could be unexpected aggregation, sedimentation, or significant parasitic effects. An example of this is shown in Figure 2, in which all mean values lie between 530 and 580 nm. Nonetheless, the highest concentration (0.5 wt %) comes with a lack of reproducibility. It should be clarified that these values for the hydrodynamic diameter do not necessarily correspond to single nanocrystals, considering their tendency to agglomerate in neutral media [14]. As long as Brownian motion prevails and there is no sedimentation, we did not attempt to disaggregate stable CNC clusters.

In the case of CNFs (Figure 2b), diluted samples (0.01 wt % or lower) presented oddly high *d_H_* values, high polydispersity (0.8–1), and a lack of reproducibility, probably due to aggregation-induced sedimentation. In contrast, a concentration of 0.05 wt % or 0.075 wt % is optimal or near-optimal for all samples, both CNFs and CNCs. Concentrations up to 0.1 wt % are also appropriate.

It should be noted that this recommended set of conditions is reported for charged nanofibers (with COO^–^ groups) and nanocrystals (with SO_3_^2–^ groups to a certain extent). It may vary for other kinds of nanocellulose. More specifically, DLS assays are not recommended for cellulosic particles with very low surface charge and *d_H_* > 100 nm, such as mechanical micro-/nanofibers [30]. In general, the consistency that is required for them to not settle over the course of the measurement lies above the maximum concentration advised by the manufacturer.

### 3.2. Dimensions of CNCs

The shape and size of CNC clusters can be appreciated from the SEM image of Figure 3 (Figure 3a). Up to 424 particles were counted from processed images (as in Figure 2b), and their mean width was collected. Its distribution, displayed as a histogram in Figure 3c, was found to be more skewed to lower sizes than DLS curves. The best fitting was that to a log-normal distribution (R^2^ = 0.97), followed by Lorentzian (R^2^ = 0.89), and lastly Gaussian (R^2^ = 0.76). The fitted log-normal function is presented in Equation (4):(4)N=kdme−(lndm170)1.9
where *N* is the expected absolute frequency, *k* is a constant that depends on the total number of counts (1.15 × 10^4^ nm^–1^ in this case), and *d_m_* is to be expressed in nanometers.

It is known that single CNCs prepared by acid hydrolysis have diameters in a much lower range, 3–50 nm, with aspect ratios between 5 and 50 [13]. In this work, the fraction between 0 and 50 nm is possibly underestimated, due to the relatively low resolution of the technique. While one of the advantages of the method is that it could be adapted to relatively low-cost tabletop microscopes, it fails to detect isolated nanocrystals whose width is less than 30 nm. Nonetheless, even DLS with backscattering, which is particularly sensitive to small particles, did not detect a significant amount of particles whose *d_H_* was lower than 30 nm.

Table 1 presents the average width from micrograph counts (237 nm) and the average hydrodynamic diameter for the concentration that yielded the highest reproducibility (0.075 wt %). This average *d_H_* value, 544 nm, is of the order of magnitude of what is commonly found in the literature by performing DLS on neutral aqueous suspensions. Some examples follow: 301 nm for CNCs from microcrystalline cellulose [31], 190 nm for CNCs from non-wood pulps, 370 nm for softwood CNCs, and 430 nm for hardwood CNCs [32].

For comparison purposes, we calculated the mean value of the maximum Feret diameter from FE-SEM images and presented it as “average length” in Table 1. The resulting value, 1290 nm, is of the same order as the estimated one, but it differs by roughly 33%. Nonetheless, length measurements from micrographs can be easily underestimated due to the rotation of CNCs or CNC aggregates inwards or outwards with respect to the plane. The fact that the calculated length matches the interval for most counts (1800–2000 nm) must be interpreted with caution. This coincidence cannot be regarded as a rigorous method validation, as neither FE-SEM nor DLS-mediated estimations are direct measurements of the length of CNC clusters in suspension. Nonetheless, the approximation can indeed be useful for comparison between different CNC samples produced by the same process.

### 3.3. Dimensions of CNFs with Different Oxidation Degrees

Width distributions for nanofibrillated oxycellulose are displayed in Figure 4. They were acceptably fitted to log-normal (R^2^ = 0.94 for 5 mmol/g, 0.93 for 10 mmol/g, and 0.96 for 15 mmol/g) or, to a lesser extent, Lorentzian functions (R^2^ = 0.87, 0.82, and 0.98, respectively). Equation (5) (CNFs5), Equation (6) (CNFs10), and Equation (7) (CNFs15) express these log-normal distribution functions:(5)n=36.8dme−(lndm185)2.4
(6)n=40.8dme−(lndm154)1.3
(7)n=36.3dme−(lndm112)1.3
where *n* is the relative frequency and *d_m_* is to be expressed in nanometers.

Table 2 shows the average width in each case, the interval with the highest frequency (mode), and the length estimated from geometrical relationships (Equation (3)). Qualitatively, the trend of CNF dimensions with the degree of oxidation was expected: the more hypochlorite, the thinner and shorter the nanofibers [33]. In quantitative terms, however, the average diameter and the corresponding aspect ratio (length/diameter) of nanofibers are below those usually reported. TEMPO-oxidized CNFs may be as thin as 5–30 nm and as slender as to have aspect ratios above 50 [22,34,35].

It can be claimed that the underestimation of the fraction of width <50 nm has a more pernicious effect in the case of nanofibers. For example, for CNFs10, the inset figure shows an isolated nanofiber with a high aspect ratio, while the average aspect ratio that can be calculated from this estimation of dimensions is as low as 14. Although very few nanofibers could be isolated in attempts to validate the estimation, the method seems more suitable for bundles and for partially fibrillated material than for isolated, slender nanofibers. In a similar approach, but using AFM instead of FE-SEM, Gamelas et al. [2] obtained diameters around 10–20 nm for TEMPO-oxidized CNFs. Given the similarity in the preparation, it may be concluded that the resolution of the microscopy technique is crucial to not overlook the most highly fibrillated fraction.

Nonetheless, despite the overestimation of the average diameter, the estimated length values, being on the micrometric scale, fit previous reports [36,37]. This is probably because both low-to-medium resolution FE-SEM and DLS with small scatter angles failed to distinguish the thinnest particles. Likewise, spaces between nanofibers in CNF bundles are ignored. Hence, the method may be reliable for CNF bundles or partially fibrillated material, but it presents limitations in what pertains to the smallest or individualized fibrils. For that, transmission electron microscopy is known to lead to better results [7].

### 3.4. Relating Key Properties of Nanocellulose to Its Dimensions

The crystallinity index found for CNCs was 95.6%, as indicated by the diffraction patterns of Figure 5. As it is well known, aqueous H_2_SO_4_ selectively hydrolyzes the least crystalline regions of cellulose, as long as its concentration is not excessive (concentrations beyond 65 wt % are not recommended for this purpose [38]).

With respect to TEMPO-mediated oxidation, the reaction is not only regioselective but also spatioselective, as the chains beneath the surface of the cellulose crystallites are not modified [36]. Therefore, the increase in the crystallinity of CNFs with the amount of hypochlorite used (Figure 5a) is solely due to the oxidative degradation of amorphous parts. Along with the oxidation of primary hydroxyls at the surface, cellulose undergoes depolymerization, particularly of its amorphous regions. This oxidative cleavage is caused not by TEMPO itself, but by the secondary oxidants, i.e., ClO^−^ and BrO^−^ [39]. In the context of nanofiber dimensions, we want to highlight that the larger the removal of amorphous parts, the shorter the fibril (Figure 5b). Selective depolymerization is one of the factors leading to smaller structures. Indeed, the degree of polymerization and the length of nanofibers are tightly correlated [40]. In any case, electrostatic repulsion and the hydration of carboxylate groups, which prevents them from hydrogen-bonding with nearby cellulose chains, are also significant in what pertains to nanocellulose dimensions.

In fact, depolymerization during TEMPO-mediated oxidation can be reliably assessed by the results from applying the Mark–Houwink equation on the intrinsic viscosity of samples [28]. Table 3 presents said results, along with those from WRV assays, potentiometric titrations, and spectrophotometry. Of those, WRV is not correlated to particle dimensions. CD, although indirectly correlated with size, is mostly due to the degree of oxidation, i.e., to the density of COO^–^ groups introduced. The low charge density of nanocrystals reveals a low degree of sulfation, which explains their tendency to aggregate.

Both DP and transmittance have a causal relationship with particle size. The former is logically one of the factors leading to size reduction. Regarding the latter, quantifying light attenuation is one of the most classical ways to estimate the size of suspended particles [41]. Here, this concept is applied to nanocellulose suspensions as a complementary and simple means to reveal trends in particle dimensions. As long as the concentration of the suspensions is held constant, the transmittance can also be used as a technique for CNC/CNF relative particle size screening. 

## 4. Conclusions

We have presented a relatively facile method for the estimation of the size of commonly prepared nanocellulose, considering the user-friendliness of recent Zetasizer models and the fact that very high-resolution microscopy is not required. In DLS assays, small scattering angles are preferred over backwards scattering, particularly for nanofibers. Likewise, a concentration of 0.05–0.10 wt % was found to be adequate in all cases, both for CNCs and CNFs.

The width distribution of CNC aggregates is skewed to smaller values and follows a log-normal pattern (R^2^ = 0.97). Geometrical relationships between this width and the *d_H_* distribution provided by DLS allow us to estimate the average length of CNC clusters as 1.9 µm. For CNFs, the results could be correlated to the ratio of oxidizing agent to pulp: 178 nm × 3.37 μm (5 mmol NaClO/g pulp), 121 nm × 1.67 μm (10 mmol NaClO/g pulp), and 92 nm × 1.23 μm (15 mmol NaClO/g pulp). Although this trend was expected, the width was probably overestimated due to the absence of single nanofibers whose diameter was lower than 30 nm.

The trend in CNF dimensions matches the trends in crystallinity, degree of polymerization, and transmittance. All considered, the estimation described here is useful for comparative purposes, for partially fibrillated materials, and for bundles or clusters of CNFs and CNCs. However, the main limitation of the method is its inapplicability to very small particle sizes (e.g., nanocrystals less than 30 nm wide), for which the resolution employed in this work is insufficient and forward scattering is not recommended.

## Figures and Tables

**Figure 1 nanomaterials-12-04288-f001:**
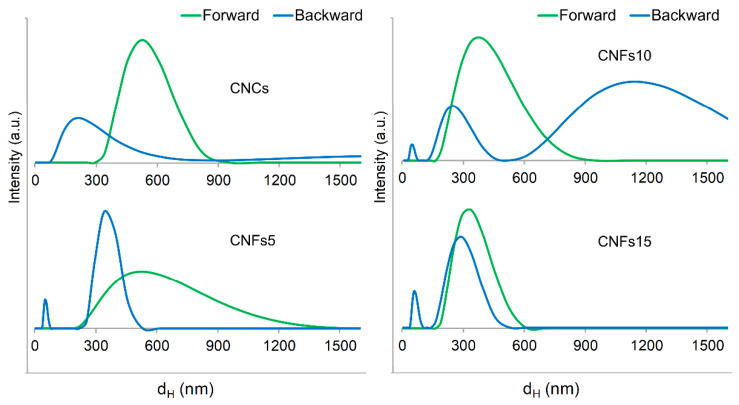
Size distribution of nanocellulose samples (consistency: 0.05 wt %), quantifying the intensity for every interval of hydrodynamic diameters (*d_H_*), setting the device either for forward angle scattering or for backscattering.

**Figure 2 nanomaterials-12-04288-f002:**
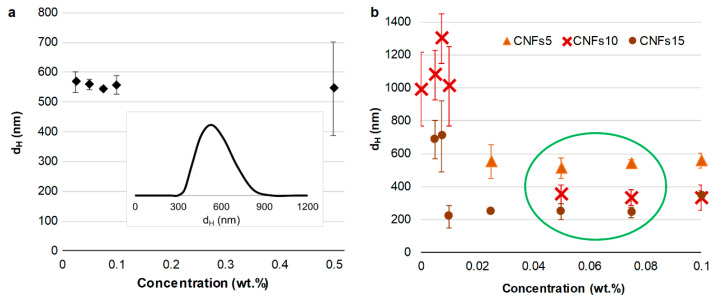
Intensity-weighted average hydrodynamic diameter of CNCs (**a**) and CNFs (**b**), as a function of concentration. The amplitude of the error bars is twice the standard deviation. Inset of (**a**) is the general shape of the size distribution (0.075 wt %).

**Figure 3 nanomaterials-12-04288-f003:**
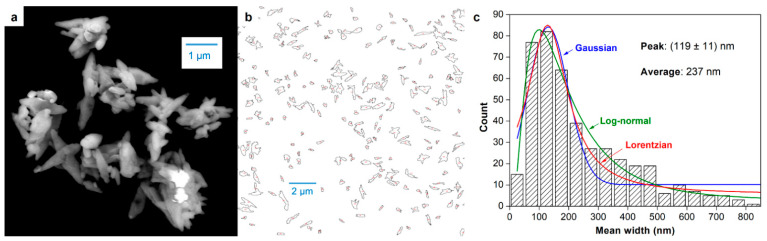
Cellulose nanocrystals and their aggregates: micrograph of a CNC cluster (**a**), outlined particles after binarization in *ImageJ* (**b**), and width histogram by counts, including fitting to Gaussian, Lorentzian, and log-normal distributions (**c**).

**Figure 4 nanomaterials-12-04288-f004:**
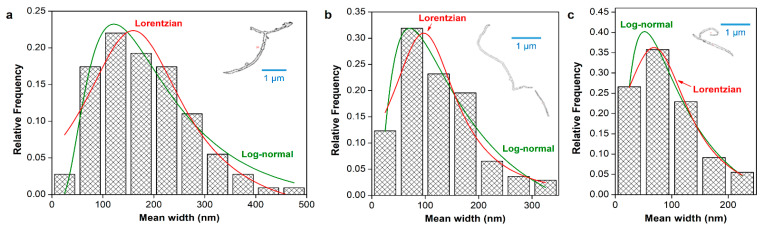
Width histograms of CNFs with different doses of hypochlorite during oxidation, namely 5 mmol/g (**a**), 10 mmol/g (**b**), and 15 mmol/g (**c**). The relative frequency is fitted to Lorentzian and log-normal distributions. Inset figures show outlined particles after binarization and isolation.

**Figure 5 nanomaterials-12-04288-f005:**
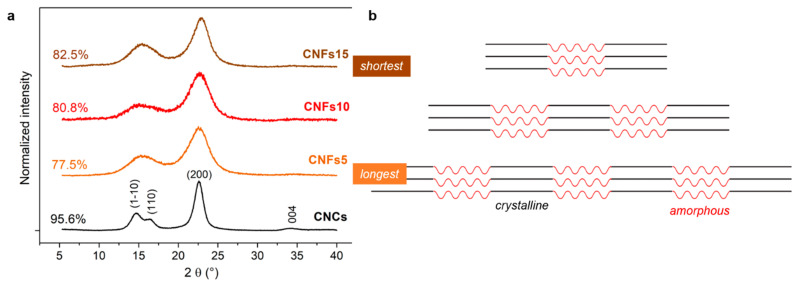
Shifted and normalized X-ray diffraction patterns of CNCs and CNFs, following a two-point baseline correction (**a**). The scheme (**b**) does not imply that depolymerization of amorphous domains is the only cause for a decrease in CNF length.

**Table 1 nanomaterials-12-04288-t001:** Estimation of the dimensions (length and diameter) of CNCs.

Dimension	FE-SEM		DLS		Calculated
	Average	Mode	Average	Mode	
Diameter (nm)	237	125–150	544 ± 6	459–571	--
Length (μm)	1.29	1.80–2.00	--		1.91

**Table 2 nanomaterials-12-04288-t002:** Estimation of the dimensions (length and diameter) of CNFs.

Sample	*d_m_* (nm)		*d_H_* (nm)		Calculated Length (μm)
	Average	Mode	Average	Mode	
CNFs5	178	100–150	543 ± 20	459–571	3.37
CNFs10	121	50–100	332 ± 49	295–342	1.67
CNFs15	92	50–100	250 ± 5	220–255	1.23

**Table 3 nanomaterials-12-04288-t003:** Key characteristics of CNCs and CNFs: degree of polymerization (DP), water retention value (WRV), charge density (CD), and transmittance at 600 nm.

Sample	DP	WRV (g/g)	CD (meq/g)	Transmittance (%)
CNCs	--	--	0.07	71.8
CNFs5	260	13.3	1.36	89.8
CNFs10	135	11.8	1.70	94.0
CNFs15	117	13.5	1.99	99.9

## Data Availability

CSV files and micrographs are available at OSF data repository: https://osf.io/b2tyq/?view_only=0bc90b82df6c487cb885d2eb777c63a7 (accessed 25 November 2022).

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
