# Peer review of "Dynamic Light Scattering Plus Scanning Electron Microscopy: Usefulness and Limitations of a Simplified Estimation of Nanocellulose Dimensions"

_nanomaterials, 2022, doi:10.3390/nano12234288_

Round 1

Reviewer 1 Report

Tarrés and coworkers provided a simple yet reliable method to estimate the dimensions of nanocellulose by using the combination of DLS and SEM. This article is well-written and well-organized. The results are trustworthy. Thus, the referee recommends that this work is acceptable in its present form.

Author Response

We are thankful to the reviewer for the positive comments.

Reviewer 2 Report

The main idea of the articles seems pretty interesting. Meanwhile, it is missing better initial materials and more microscopies to assume that DLS will really work. For example, with the SEM images for the produced nanocrystals it is not possible to measure the lengh and diameter accordingly because there are clusters but not isolated nanocrystals. Also the images for all the nanofibers produced are not shown so it is not possible to see which kind of fibers were obtained, how was the dimensions, nothing. The main idea from the article is good but it is necessary more information to validate such hypothesis using more techniques as well like AFM, SEM and TEM.

Author Response

We are thankful for this opportunity to improve our manuscript. Hopefully, this response will cover all of the issues raised by the reviewer:

  • Untreated micrographs have been made available, along with the CSV files for size distributions + XRD patterns, at OSF repository. The link is found at the end of the manuscript. Short version: https://osf.io/b2tyq/  — Please see the attached PDF.
  • Along the manuscript, we insist on the limitations of the method, as the output describes clusters/bundles. This has been enriched in the revised version. For instance, line 291: 

Hence, the method may be reliable for CNF bundles or partially fibrillated material, but it presents limitations in what pertains to the smallest or individualized fibrils. For that, transmission electron microscopy is known to lead to better results [7].

  • Regarding the materials, DLS would probably not work for uncharged nano-/microcelluloses, unless they are really small in size (dH < 100 nm). On one hand, these nanofibers and nanocrystals with significant surface charge did not settle during DLS assays. On the other, limitations mostly come from SEM. E.g. lines 202 and following.
  • SEM was chosen in this work with a purpose of differentiation in mind, as the dimensions of nanofibers have been estimated by AFM in Ref. 2 (Gamelas et al.) and, even more accurately, by TEM in Ref. 7 (Campano et al.). The latter group of authors can rely on TEM + image treatment software without additional techniques. Nonetheless, many recent tabletop SEM devices are much more affordable and user-friendly than usual TEM, and thus we believe that many other groups will welcome our study — even if they end up discarding the possibility once they find out the limitations.

Reviewer 3 Report

Comments to the Author

The authors develop a very useful method to determine the size of nanocellulose. The results are validated experimentally. The paper is very good (perhaps even excellent) and can be accepted in its present form. I have just one minor question about the desperation on the experiments.

1. In the description of Figure 5, the authors declare that the larger the removal of amorphous parts, the shorter the fibril. My question is how to quantify the length of the amorphous portion of cellulose?

Author Response

The authors develop a very useful method to determine the size of nanocellulose. The results are validated experimentally. The paper is very good (perhaps even excellent) and can be accepted in its present form. I have just one minor question about the desperation on the experiments.

  1. In the description of Figure 5, the authors declare that the larger the removal of amorphous parts, the shorter the fibril. My question is how to quantify the length of the amorphous portion of cellulose

We are grateful to the reviewer for the positive comments and for the pertinent question.

The size of crystallites can be estimated from XRD patterns (included among the CSV files) by using Scherrer's equation, and it would yield approximately 1.8 nm. Still, other than describing the proportion of amorphous domains as roughly 20% along the fibril, our groups lack means to fathom how long each amorphous domain is. The depictions of Figure 5 are illustrative, as mapping cellulose fibrils (or, on top of that, with oxycellulose on their surface) is out of the scope of the paper.

Mapping the fibril, in any case, is not required to sustain the claim, and this is why: if there is hydrolysis, and this is known to happen preferably on amorphous regions, there is a loss in DP and subsequent shortening. Chain length and fibril length are not the same, but they are so closely correlated — see, e.g., Shinoda, Saito, Okita, Isogai, Relationship between Length and Degree of Polymerization of TEMPO-Oxidized Cellulose Nanofibrils, Biomacromolecules (2012). The revised version has been enriched to clarify these points.

Reviewer 4 Report

In this paper, the usefulness and limitations of DLS and SEM for the measurement of nanocellulose dimensions were discussed. And the forward scattering (12.8º) and concentrations around 0.05-0.075% of nanocellulose dispersion were recommended for DLS. However, this concentration range is very narrow.

1) Only CNC prepared by sulfuric acid and CNF prepared from TEMPO oxidation were estimated using DLS and SEM. How about the nanocellulose prepared by other methods (e.g. HCl hydrolysis, formic acid hydrolysis, mechanical methods)?

2) SEM cannot give accurate size of nanocellulose, because samples before measurement should be dried and coated with platinum or carbon. So the size with SEM was over-evaluated. Also, the samples were aggregated and self-assembled after drying. Why don’t you use TEM compared to DLS?

Author Response

Note: The attachment contains the same response, but it may be easier to follow.

In this paper, the usefulness and limitations of DLS and SEM for the measurement of nanocellulose dimensions were discussed. And the forward scattering (12.8º) and concentrations around 0.05-0.075% of nanocellulose dispersion were recommended for DLS. However, this concentration range is very narrow.

Response to general comments. We acknowledge the reviewer’s understanding of the manuscript. While we do not recommend concentrations below 0.05%, the upper end can be safely increased to 0.10%. The increase in deviation is little or, in most cases, non-significant. It is not a huge improvement but, in the abstract and the conclusions of the revised version, we state: “0.05 – 0.10 wt.%”.

We have also changed the wording in the corresponding section (~ line 203):

a concentration of 0.05 wt.% or 0.075 wt.% is appropriate for all samples, both CNFs and CNCs à a concentration of 0.05 wt.% or 0.075 wt.% is optimal or near-optimal for all samples, both CNFs and CNCs. Concentrations up to 0.1 wt.% are also appropriate.

Point 1. Only CNC prepared by sulfuric acid and CNF prepared from TEMPO oxidation were estimated using DLS and SEM. How about the nanocellulose prepared by other methods (e.g. HCl hydrolysis, formic acid hydrolysis, mechanical methods)?

Response to point 1. We would like to argue that the use of sulfuric acid is the most common way to prepare CNCs, while the generation of carboxylate groups is likely the most usual method towards “true” nanofibers — in the sense that fully mechanical methods tend to yield microfibers. This is a pertinent question because while the systematic error by SEM would be lower, micro-/nanofibers whose Z-potential is not high enough tend to aggregate, and, as such, they are not suitable for DLS measurements. Likewise, CNCs with no charged groups (sulfate or any other kind) could face a similar problem. Hence, in the revised version, we refer to the limitations of uncharged nanocellulose:

It should be noted that this recommended set of conditions is reported for charged nanofibers (with COO groups) and nanocrystals (with SO32– groups to a certain extent). It may vary for other kinds of nanocellulose. More specifically, DLS assays are not recommended for cellulosic particles with very low surface charge and dH > 100 nm, such as mechanical micro-/nanofibers [31]. In general, the consistency that is required for they not to settle over the course of the measurement lies above the maximum concentration advised by the manufacturer.

Ref. 31 refers to an article from researchers from Medellin & Grenoble on the dimensions and other characteristics of various mechanical nanocelluloses.

Point 2. SEM cannot give accurate size of nanocellulose, because samples before measurement should be dried and coated with platinum or carbon. So the size with SEM was over-evaluated. Also, the samples were aggregated and self-assembled after drying. Why don’t you use TEM compared to DLS?

Response to point 2. There is no doubt that TEM (Campano et al., ref. 7) outperforms SEM in this regard, and it probably outperforms AFM too. One of the reasons for choosing SEM was to differentiate ourselves from previous works. The other reason is the different availability and skill requirement of SEM vs. TEM, which have been accentuated over the last few years. There are many new tabletop SEM devices that can be found at affordable prices and require minimum configuration or conditioning. Thence we believe that many research groups will be interested, not only in these possibilities of SEM, but also in the important limitations that we (and the reviewer) indicate. Line 291: “Hence, the method may be reliable for CNF bundles or partially fibrillated material, but it presents limitations in what pertains to the smallest or individualized fibrils.”  Addition to the revised version: For that, transmission electron microscopy is known to lead to better results [7].

Round 2

Reviewer 4 Report

To me, the current version can be accepted.